# Noncentrosymmetric Supramolecular Hydrogen-Bonded Assemblies Based on Achiral Pyrazine-Bridged Zinc(II) Coordination Polymers with Pyrazinedione Derivatives

Ko Yoneda [1], Ken Kanazashi [2], Hitoshi Kumagai [3,†], Ryuta Ishikawa [2,*] and Satoshi Kawata [2,*]

1    Department of Chemistry and Applied Chemistry, Faculty of Science and Engineering, Saga University, 1 Honjo-machi, Saga 840-8502, Japan
2    Department of Chemistry, Faculty of Science, Fukuoka University, 8-19-1 Nanakuma, Jonan-ku, Fukuoka 814-0810, Japan
3    Institute for Molecular Science (IMS), 38 Nishigonaka, Myodaiji-cho, Okazaki 444-8585, Japan
*    Correspondence: ryutaishikawa@fukuoka-u.ac.jp (R.I.); kawata@fukuoka-u.ac.jp (S.K.)
†    Present address: Toyota Central R and D Laboratories, Inc., Yokomichi, Nagakute 470-1192, Japan.

**Abstract:** Reaction of $M(OAc)_2 \cdot xH_2O$ (M, $x$ = Zn, 2 and Co, 4), 1,4-dihydro-5,6-dicyano-2,3-pyrazinedione ($H_2CN_2pyzdione$), and pyrazine (pyz) affords two compounds of the same molecular formula $\{[M(H_2O)_6][M(CN_2pyzdione)_2(pyz)]\cdot 6H_2O\}_n$ (M = Zn for **1** and Co for **2**) in which discrete units of $[M(H_2O)_6]^{2+}$ are linked to one-dimensional chains of $[M(CN_2pyzdione)_2(pyz)]^{2-}$ via multiple O–H⋯O hydrogen-bonding interactions and $M^{2+}$-bound $H_2O$ molecules in $[M(H_2O)_6]^{2+}$ also serve as linkers of hydrogen-bonded interstitial $H_2O$ molecules. Remarkably, **1** crystallizes in the monoclinic crystal system, the similar crystal system and unit cell parameters as **2**, but with a space group distinct from **1** and **2**, i.e., **1** is the noncentrosymmetric space group $C2$, whereas **2** is the centrosymmetric space group $C2/m$. This polar structure for **1** is induced by the presence of alternating arrangements of distinguishable two axial Zn–N bonds within $[Zn(CN_2pyzdione)_2(pyz)]^{2-}$ chains. Indeed, solid-state circular dichroism spectra of **1** exhibit significant Cotton effects, as evidenced by the polar space group $C2$. Moreover, these Cotton effects show clear temperature-dependence depending on contents of $H_2O$ molecules of **1**.

**Keywords:** chiral structure; hydrogen bonding assembly; zinc ion; cobalt ion

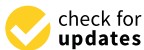



The syntheses of coordination polymers (CPs) with specific network topology are attracting much interest due to not only their intriguing structural diversity of the extended lattices but also for chemical and physical properties such as catalysis, gas sorption, magnetic and conductive properties [1–3]. These materials are synthesized using the concept of crystal engineering or supramolecular engineering to control the topology and geometry of the network formed. Polydentate ligands, such as polycyanides, polycarboxylates, polypyridines and poly-amines have so far been used in constructing extended network materials [1–6]. In this concern, 1,4,5,6-tetrahydro-5,6-dioxo-2,3-pyrzinedicarbonitrile ($H_2CN_2pyzdione$) has attracted our interest as a multiconnecting ligand. The ligand has two types of protonation sites to give 2,3-pyrazinedione form or 2,3-pyrazinediol form. (Scheme 1). We have been working on $H_2CN_2pyzdione$ to develop coordination- and hydrogen-bond supported metal–$CN_2pyzdione^{2-}$ coordination materials and have reported topological and physico-chemical properties [7–9].

(a)                              (b)

**Scheme 1.** (**a**) 2,3-pyrazinediol form, (**b**) 2,3-pyradinedione form.

Here, we encountered a chiral CP with use of pyrazine (pyz) as co-bridging ligands with divalent transition metal ions. Chirality has been an important topic not only in chemistry and biochemistry but also in materials science [10–14]. Chiral compounds have been synthesized by two routes. One is the synthesis using chiral reactants and the other is spontaneous resolution without any chiral reagents [14]. While chiral CPs become a topic of intense interest due to their intriguing potential applications, the mechanism of spontaneous resolution and physical theory are not yet fully understood [12–14]. Here, we report synthesis, single crystal structures of a chiral and an achiral CP and their thermal and optical properties. Chiral or achiral structures were controlled by changing metal ions.

The synthesis involves the deprotonation of $H_2CN_2$pyzdione with $Zn(OAc)_2 \cdot 2H_2O$ or $Co(OAc)_2 \cdot 4H_2O$ and formation of corresponding aquo complex, $[M(H_2O)_6]^{2+}$ (M = Zn or Co) in $EtOH/H_2O$ mixed solvent followed by in situ self-assembly reactions with pyz as bridging ligands (Experimental Section, Supplementary Materials). Colorless or orange crystals suitable for X-ray structure analysis were obtained for **1** and **2**, respectively. The both compounds exhibit the same chemical formula, $\{[M(H_2O)_6][M(CN_2pyzdione)_2(pyz)] \cdot 6H_2O\}_n$. Moreover, powder X-ray diffraction (PXRD) patterns for finely ground microcrystalline samples of **1** and **2** matched well with their calculated patterns simulated from single crystal diffraction data, confirming no crystal polymorphism. The single crystal structure analysis of **1** reveals that the compound crystallizes in the monoclinic space group *C*2 with an absolute structure (Flack) parameter of 0.019(5), indicating a polar structure despite of the use of achiral starting materials.

The crystal structure of **1** consists of crystallographically independent polymeric structures, discrete $[Zn(H_2O)_6]^{2+}$ units and uncoordinated $H_2O$ molecules packed together (Figure S1, Supplementary Materials). In total, **1** can be formulated as $\{[Zn(H_2O)_6][Zn(CN_2pyzdione)_2(pyz)] \cdot 6(H_2O)\}_n$ and the oxidation states of the both Zn centers were established to be 2+ by bond valence sum calculations and charge balance considerations.

Two distinguishable C1–C2 and C3–C4 distances of 1.490(2) and 1.374(3) Å within the $CN_2pyzdione^{2-}$ ligand are of single and double bond, respectively, while respective average C–N of the ring and C–O distances are 1.350(2) and 1.283(2) Å, respectively. These bond distances essentially suggest a 2,3-pyrazinedione dianionic form, namely, an oxamide form rather than as a 2,3-pyrazinediolato form [15], which are similar to those of previously reported $CN_2pyzdione^{2-}$-bound coordination compounds [16].

The polymeric structure is an anionic one-dimensional (1D) array consists of Zn1 centers decorated by symmetry-related planar chelating $CN_2pyzdione^{2-}$ ligands bridged by pyz ligands (Figure 1). In the 1D array, the Zn1 centers and N5 and N6 atoms of the pyz ligands lie on a crystallographic two-fold rotation axis of a polar space group *C*2, thus, the $\cdots N5–Zn1–N6^i\cdots$ sequence make a straight chain propagating along the crystallographic *b*-axis. The 1D chains are homochiral with the two-fold axis passing through the $\cdots N5–Zn1–N6^i\cdots$ connectivity. The intrachain $Zn1\cdots Zn1^{ii}$ separation through the pyz bridge is 7.0797(5) Å. $H_2CN_2$pyzdione ligand has two protons and two possible protonation sites. In the present case, the ligands are deprotonated to yield dianions to balance the charges of the cations. The planar $CN_2pyzdione^{2-}$ ligands of the chain form interchain stacks in the direction of the *b*-axis and the shortest inter ligand $C3\cdots C3$ * (* symmetry code = $-x + 1/2, y-1/2, -z + 1$) distance of 3.579(3) Å is indicative of the presence of some degree of π–π stacking interaction to give a two-dimensional (2D) network spreading out along the *bc*-plane. Zn1 centers of the anionic chains display six-coordinate $\{ZnN_2O_4\}$

octahedral environments, consisting of two N atoms of pyz in *trans* positions and four O atoms of chelating $CN_2$pyzdione$^{2-}$ ligands in the basal plane. The distortion of the octahedral geometry is characterized by O1–Zn1–O2 bond angles (79.34(4)°) and slightly long Zn1–N5 and Zn1–N6 bond distances (2.146(3) and 2.159(3) Å) compared to those of Zn1–O1 and Zn1–O2 (2.1269(11) and 2.0913(12) Å) are indicative of the elongation axis is N5–Zn1–N6$^i$ which runs the chain direction. It is worth noting in the Zn–N$_{pyz}$ distances that there are long Zn1–N5 (2.146(3)) and short Zn1–N6 (2.159(3) Å) bond distances. Since the compound **1** belongs to a chiral space group, the short and long Zn–N$_{pyz}$ bonds alternatively aligned parallel to the crystallographic *b*-axis. The Zn2 center of the discrete $[Zn(H_2O)_6]^{2+}$ unit also sits on the crystallographic two-fold rotation axis and aligned along crystallographic *b*-axis. The Zn2 center has slightly elongated octahedral coordination geometry, with the three Zn2–O3, Zn2–O4, and Zn2–O5 bond distances of 2.0783(13), 2.0970(17), 2.0762(17) Å. The O–Zn2–O$^{iv}$ bond angles are in the range of 88.16(8)–92.27(9)°. The bond distances and angles are similar to those found in the Zn$^{II}$-type Mohr salt (known more commonly as Tutton salt) of $(NH_4)_2[Zn(H_2O)_6](SO_4)_2$ [17]. In the structure, the octahedron shows tilting and cations are interlayered by hydrogen-bonding interactions within the 2D network formed by the π–π stacking of the anionic chains. The structure exhibits several hydrogen-bonding networks in the crystal packing (Figure 2). First, hydrogen-bonding interaction is O–H···O hydrogen-bonding interactions, which is found in between $[Zn(CN_2pyzdione)_2(pyz)]^{2-}$ and $[Zn(H_2O)_6]^{2+}$ units. The hydrogen-bonding donors (HD) and acceptors (HA) are the coordinated $H_2O$ molecules and oxygen atoms of $CN_2$pyzdione$^{2-}$ ligands, respectively (Figure 2a). Second, hydrogen-bonding interaction is O–H···O hydrogen-bonding interactions between oxygen atoms of $[Zn(H_2O)_6]^{2+}$ units and oxygen atoms of lattice $H_2O$ solvents (Figure 2b). Third, hydrogen-bonding interaction is O–H···O hydrogen-bonding interactions between lattice $H_2O$ molecules (Figure 2b). All hydrogen-bonding distances are shown in Table S4 (Supplementary Materials). The characteristic supramolecular structure of this compound is constructed by means of the layered structure of the π-stacked layers of $[Zn(CN_2pyzdione)_2(pyz)]^{2-}$ chains in the *bc*-plane separated by the hydrogen-bonded layers of $[Zn(H_2O)_6]^{2+}$ units aligned along *b*-axis and interstitial $H_2O$ molecules. These hydrogen-bonding, π–π stacking and electrostatic interactions result in the formation of a 3D supramolecular structure. The interlayer distance defined by Zn1···Zn1 * (* symmetry code = *x*, *y*, *z* + 1) separation is 10.0831(5) Å. This structural feature resembles those found in the crystal structures of metal assembled systems {(Hphz) (H$_{0.5}$phz)$_2$ [Fe(CA)$_2$(H$_2$O)$_2$](H$_2$O)$_2$}$_n$ (Hphz = phenazinium, CA$^{2-}$ = chloranilate) [18], {(3-pyOH$_2$)$_2$[Co(CA)$_2$(H$_2$O)$_2$]}$_n$ (3-pyOH$_2$ = 3-hydroxypyridinium) [19] and [Co(H$_2$O)$_6$](H$_2$pm) (H$_2$pm$^{2-}$ = pyromellitate) [20], in which hydrogen-bonding interactions and electrostatic interactions between the ions stabilize the supramolecular structures.

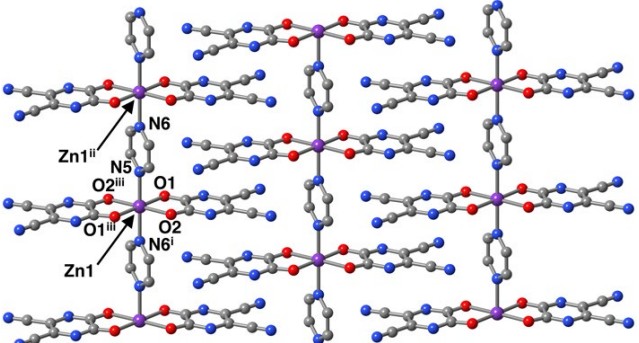

**Figure 1.** View of the arrangement of 1D chains of $[Zn(CN_2pyzdione)_2(pyz)]^{2-}$ in the crystal packing of **1** with selective atom labeling, where purple, grey, blue, and red spheres represent Zn, C, N, and O atoms, respectively. H atoms have been omitted for clarity. i–iii denote symmetry codes: (i) *x*, *y*–1, *z* (ii) *x*, *y* + 1, *z*, and (iii) –*x* + 1, *y*, –*z* + 1.

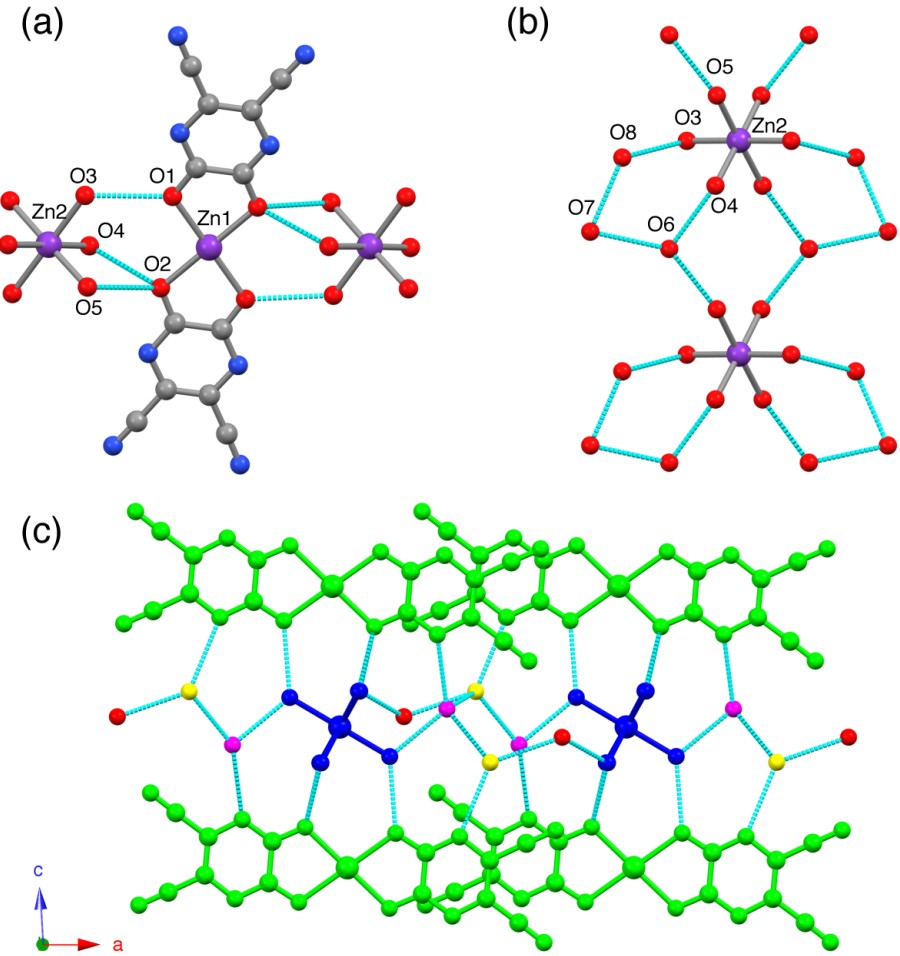

**Figure 2.** Crystal structure for **1**, highlighting hydrogen-bonding networks (broken cyan lines). (**a**) Hydrogen-bonding interactions between $[Zn(CN_2pyzdione)_2(pyz)]^{2-}$ and $[Zn(H_2O)_6]^{2+}$ units. (**b**) Hydrogen-bonding interactions between $[Zn(H_2O)_6]^{2+}$ units and lattice $H_2O$ molecules, and between lattice $H_2O$ molecules. (**c**) View down of the *b*-axis, where green parts represent $[Zn(CN_2pyzdione)_2(pyz)]^{2-}$ units, blue parts represent $[Zn(H_2O)_6]^{2+}$ units, and magenta, red, and yellow balls represent O atoms of lattice $H_2O$ molecules, respectively. H atoms are omitted for clarity.

Structure of **2** is composed also of polymeric 1D chains $[Co(CN_2pyzdione)_2(pyz)]^{2-}$, discrete $[Co(H_2O)_6]^{2+}$ units and uncoordinated $H_2O$ molecules and the supramolecular structure is constructed by $\pi$-stacked layers of anionic $[Co(CN_2pyzdione)_2(pyz)]^{2-}$ chains and the hydrogen-bonded layers of $[Co(H_2O)_6]^{2+}$ units and interstitial $H_2O$ molecules (Figure S2, Supplementary Materials). In contrast to that of **1**, **2** crystallized in an achiral space group, $C2/m$ and both Co1 and Co2 centers are located on the crystallographic two-fold rotation axis and mirror plane, consequently **2** exhibits a more symmetric structure than that of **1**. While **1** has long and short Zn1–N5 and Zn1–N6 bond distances, which alternatively aligned in the 1D chain, the Co1–N5 distances are symmetry-related and, thus, they are equivalent in **2**. Reflecting the crystallographic symmetry between **1** and **2**, the discrete $[Co(H_2O)_6]^{2+}$ unit also shows more symmetric structure with two Co2–O3 and Co2–O4 bond distances of 2.063(2) Å and 2.0915(16) Å (vide supra, there are three Zn2–O3, Zn2–O4, and Zn2–O5 bonds for **1**). Coordinated $H_2O$ molecules form asymmetric hydrogen bonds in **1**. On the other hand, the two $H_2O$ molecules are symmetry-related so that the hydrogen bond distances are equal in **2**, which also contains interstitial $H_2O$ molecules similar to that of **1**. However, the interstitial $H_2O$ molecules (O6 and O7) are disordered over two symmetry-related sites with 50% occupation to yield symmetric hydrogen-bonding interactions with nitrogen atoms of 1D chain. In contrast to the chiral

structure of **1**, in which $H_2O$ molecules are crystallographically ordered, interstitial $H_2O$ molecules are symmetrically disordered to give an achiral structure of **2**. These structural differences are indicative of an important role of the interstitial $H_2O$ molecules and chirality of the crystal structures.

The thermal property of **1** was examined by thermogravimetric analysis (TGA) (Figure S3, Supplementary Materials). They are characterized by three weight loss steps in the range 300–450 K at heating rate of 10 K/min. The first weight loss step exhibit rapid weight loss of 14.4% up to 320 K and the second weight loss step shows slow weight loss of 4.8% from 320 K up to 330 K, which corresponds to six interstitial $H_2O$ molecules and two coordinated $H_2O$ molecules to give $\{[Zn(H_2O)_6][Zn(CN_2pyzdione)_2(pyz)]\}_n$ phase and $\{[Zn(H_2O)_4][Zn(CN_2pyzdione)_2(pyz)]\}_n$ phase, respectively. The third weight loss of 4.8% from 360 K is due to the loss of the two coordinated $H_2O$ molecules to yield $\{[Zn(H_2O)_2][Zn(CN_2pyzdione)_2(pyz)]\}_n$ phase. Variable-temperature powder X-ray diffraction (PXRD) was measured to see the structural properties. The results are shown in Figure S4 (Supplementary Materials). Broad and weak intensities of the diffraction peaks after dehydration are attributed to a decrease in correlation length of the structural ordering. The diffraction pattern at 310 K is similar to that of the virgin form indicating that the structure is not globally changed, and structure is stable up to 310 K. The sample heated to 315 K showed diffraction peaks of the virgin form and new diffraction peaks, indicating the mixture of the virgin form and partially dehydrated form. The sample was heated to 320 K to complete the liberation of interstitial $H_2O$ molecules, the diffraction peaks of the virgin form were disappeared, indicating the structural change. With increasing temperature to 360 K to give $\{[Zn(H_2O)_2][Zn(CN_2pyzdione)_2(pyz)]\}_n$ phase, diffraction pattern exhibits different diffraction pattern from that of $\{[Zn(H_2O)_6][Zn(CN_2pyzdione)_2(pyz)]\}_n$ at 320 K. The sample was further heated from 360 K to 420 K; severe loss of intensity and broadening of the diffraction peaks are indicative of the low crystallinity of $\{[Zn(H_2O)_2][Zn(CN_2pyzdione)_2(pyz)]\}_n$ phase. Figure 3 shows the temperature dependence of the intensities of diffraction peaks observed at $2\theta = 8°$ of the virgin form and $2\theta = 13°$ of $\{[Zn(H_2O)_4][Zn(CN_2pyzdione)_2(pyz)]\}_n$ phase. The intensity of the diffraction peak at $2\theta = 8°$ rapidly decreased at 315 K. This is accounted for by liberation of interstitial $H_2O$ molecules to yield $\{[Zn(H_2O)_6][Zn(CN_2pyzdione)_2(pyz)]\}_n$ phase. On the other hand, the intensity of the diffraction peak at $2\theta = 13°$ gradually increased with increasing temperature from 320 K and decreased with increasing temperature from 350 K. While the weight of the sample is almost unchanged, indicating the liberation of the $H_2O$ molecules is almost finished in the temperature range 330 K to 360 K, change in the intensity of the diffraction peak is indicative of solid-state structural transformation. We have reported solid-state structural transformation in a similar $\{[Zn(pyz)(H_2O)_4][Zn(CN_2pyzdione)_2(pyz)]\}_n$ system in which liberation of coordinated $H_2O$ molecules have triggered the solid-state reaction [7]. Removal of the coordinated $H_2O$ molecules by heat treatment yields $\{[Zn(pyz)(H_2O)_2][Zn(CN_2pyzdione)_2(pyz)]\}_n$ phase, in which open coordination sites were created to react in the solid state and the vacant sites are occupied by the neighboring ligand. To examine the solid-state structural transformation in **1**, differential scanning calorimetry (DSC) measurement was performed (Figure S5, Supplementary Materials). DSC trace of **1** shows thermal anomaly at 340 K, indicating the occurrence of a structural transformation in the solid state. Broad and weak intensities of diffraction pattern prevent Rietveld refinements at this stage. We assumed that the oxygen atom of $CN_2pyzdione^{2-}$ ligands in the cationic 1D chain occupied the open coordination sites of $\{[Zn(H_2O)_4][Zn(CN_2pyzdione)_2(pyz)]\}_n$ phase similar to previously reported $\{[Zn(pyz)(H_2O)_2][Zn(CN_2pyzdione)_2(pyz)]\}_n$ phase.

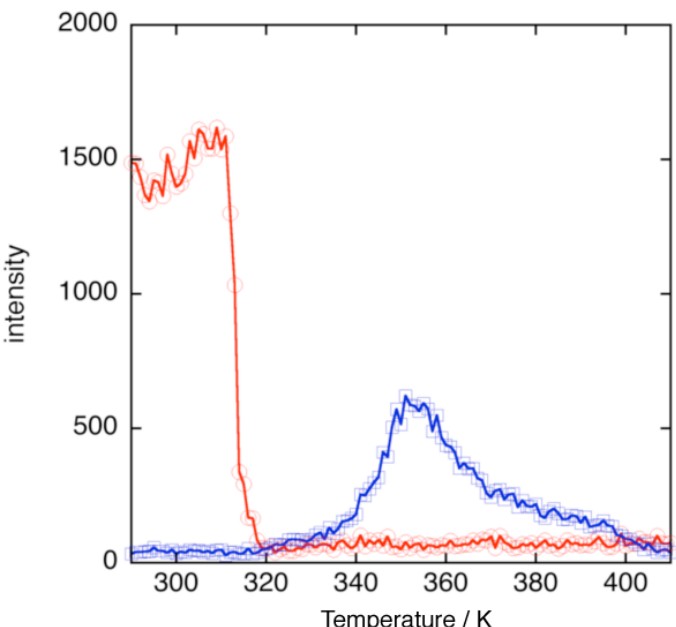

**Figure 3.** Temperature dependence of diffraction intensities at $2\theta = 8°$ (red) and $13°$ (blue).

Based on the structural characterization, we conducted circular dichroism (CD) and diffuse reflectance spectroscopy in the solid-state to identify whether the bulk crystals are racemic or enantiomeric excess for **1**. Bulk crystals were carefully ground along the crystal facet, and CD spectra were measured. Figure S6 shows the diffuse reflectance spectrum and CD spectra. The spectrum showed absorption bands at 250 nm, 300 nm and shoulder at 400 nm. These bands are thought to be charge transfer (CT) bands. The CD spectra exhibit peaks at 250, 260 and 340 nm as shown in Figure S6 (Supplementary Materials), indicating that the bulk crystals of compound is enantiomeric excess rather than racemic. It is worthwhile to examine the temperature dependence of CD spectroscopy in the dehydration process. Chiral CPs have been reported to date; however, the reports on the electronic diffuse reflectance spectroscopy and CD spectroscopy during the dehydration process in the chiral systems are still sparse. The absorption at around 400 nm in the electronic reflectance spectrum increased at 343 K. Based on the results of TGA measurements, the absorption band comes from $\{[Zn(H_2O)_4][Zn(CN_2pyzdione)_2(pyz)]\}_n$ phase. Variable temperature dependence of CD spectrum is shown in Figure S6 (Supplementary Materials) and plot of the temperature dependence of intensities of 250 nm, 270 nm and 400 nm is shown in Figure 4. The CD ellipticity at 400 nm increased at 325 K and decreased at 340 K. These results are consistent with TGA, XRD and DSC measurements, which liberation of coordinated $H_2O$ molecules started to give $\{[Zn(H_2O)_4][Zn(CN_2pyzdione)_2(pyz)]\}_n$ phase at 325 K and the solid-state structural transformation started at 340 K. These results indicate that the compound retains the chirality during dehydration and after solid-state structural transformation.

In summary, a new chiral and an achiral CP have been synthesized in the reactions of M(II) ion (M = Zn (**1**) or Co (**2**)), $H_2CN_2pyzdione$ and pyz. Single crystal structure analysis of **1** revealed that **1** belonged to a chiral space group, $C2$ and was formulated as $\{[Zn(H_2O)_6][Zn(CN_2pyzdione)_2(pyz)]\cdot6(H_2O)\}_n$, in which anionic 1D chains formed 2D network by $\pi$-$\pi$ stacking interactions and $[Zn(H_2O)_6]$ units and $H_2O$ molecules were hydrogen bonded. Chirality is retained by **1** during the dehydration process as indicated by variable temperature CD spectrum measurements in the solid state. Our results herein reported are indicative of the important role of interstitial $H_2O$ molecules to a chiral structure.

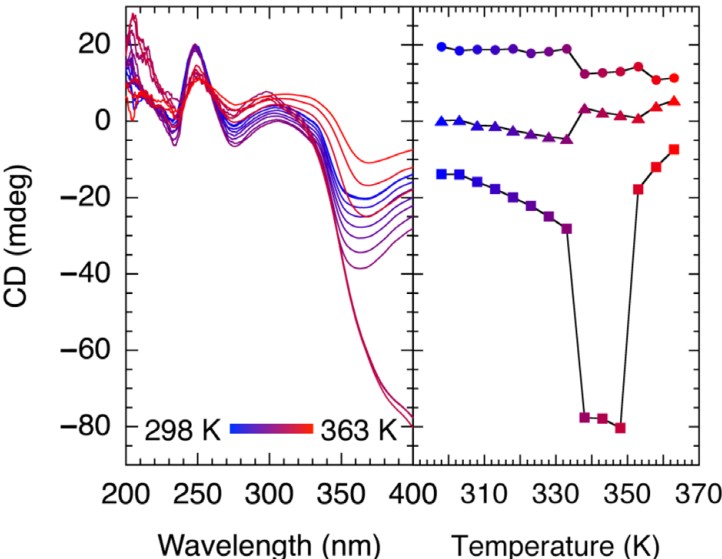

**Figure 4.** Left: variable temperature CD spectra at 5 K intervals in the temperature range 298–363 K of **1**. Right: temperature dependence of CD ellipticities at 250 (circles), 270 (triangles), and 400 (squares) nm, respectively, for **1**.

**Supplementary Materials:** The following supporting information can be downloaded at: https://www.mdpi.com/article/10.3390/chemistry5010015/s1, Experimental Section; Figure S1: Coordination around the zinc atoms and the adopted atom numbering for 1; Figure S2: Coordination around the cobalt atoms and the adopted atom numbering for 2; Figure S3: TGA and differential thermal analysis (DTA) data for **1**; Figure S4: In situ variable temperature PXRD patterns (left) at given temperatures and TGA data (right) for **1**; Figure S5: Results of DSC measurement and time derivative of DSC.; Figure S6: Solid state diffuse reflectance (green solid line) and CD (red solid line) spectra of **1** in the UV region at room temperature.; Table S1: Summary of crystal data for **1** and **2**; Table S2: Selected bond distances and angles for **1**; Table S3: Selected bond distances and angles for **2**; Table S4: Hydrogen-bond distances (Å) for **1**; Table S5: Hydrogen-bond distances (Å) for **2**.

**Author Contributions:** Conceptualization, S.K. and R.I.; methodology, S.K. and R.I.; validation, S.K. and R.I.; formal analysis, R.I., S.K. and K.K.; investigation, K.K. and H.K.; resources, R.I.; data curation, R.I. and K.K.; writing—original draft preparation, K.Y.; writing—review and editing, S.K. and R.I.; visualization, R.I.; supervision, S.K.; project administration, S.K. and R.I.; funding acquisition, S.K. and R.I. All authors have read and agreed to the published version of the manuscript.

**Funding:** This research was funded by JSPS KAKENHI (grant number 20K05546) and Fukuoka University (grant number 205002). The APC was funded by MDPI Chemistry.

**Institutional Review Board Statement:** Not applicable.

**Informed Consent Statement:** Not applicable.

**Data Availability Statement:** The data presented in this study are available on request from the corresponding author.

**Conflicts of Interest:** The authors declare no conflict of interest.

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
