# Peer review of "Noncentrosymmetric Supramolecular Hydrogen-Bonded Assemblies Based on Achiral Pyrazine-Bridged Zinc(II) Coordination Polymers with Pyrazinedione Derivatives"

_chemistry, doi:10.3390/chemistry5010015_

Round 1

Reviewer 1 Report

Review reports should contain the following:

 A brief summary (one short paragraph) outlining the aim of the paper, its main contributions and strengths.

The present paper continues the authors’ series of works regarding the synthesis of coordination polymers (CPs) and their topological and physico-chemical properties. The authors synthesized CPs in the reactions of Zn(II) or Co(II) ions with H2CN2pyzdione and pyrazine.

Generalconceptcomments. Article: highlighting areas of weakness, the testability of the hypothesis, methodological inaccuracies, missing controls, etc. These comments are focused on the scientific content of the manuscript and should be specific enough for the authors to be able to respond.

No specific areas of weakness regarding scientific content have been found.

Is the manuscript clear, relevant for the field and presented in a well-structured manner?

The manuscript is relevant for the field, clearly written and well-structured.

Are the cited references mostly recent publications (within the last 5 years) and relevant? Doesitincludeanexcessivenumberofself-citations?

All the references cited in the manuscript are relevant with more than half of them being published recently. The number of self-citing publications is appropriate, no excess is observed.

Is the manuscript scientifically sound and is the experimental design appropriate to test the hypothesis?

The manuscript is scientifically sound and the used methods are appropriate for the set goals.

Are the manuscript’s results reproducible based on the details given in the methods section?

The authors do not provide methods of synthesis and purification of target CPs either in the article or in the Supporting Information. It is therefore not possible to assess reproducibility of the results.

Are the conclusions consistent with the evidence and arguments presented?

The conclusions are consistent with the presented data and arguments.

Specific comments referring to line numbers, tables or figures that point out inaccuracies within the text or sentences that are unclear. These comments should also focus on the scientific content and not on spelling, formatting or English language problems, as these can be addressed at a later stage by our internal staff.

Page 4, line 160: Captions to Figure 2 and Figure 3 overlap.

Page 6, lines 221–223: the caption to Figure 4 is located both below and above the figure.

Author Response

Dear Dr. Yu and reviewers,

  On behalf of my co-authors, I appreciate you and the reviewers for the positive and constructive comments and suggestions on the manuscript entitled “Noncentrosymmetric Supramolecular Hydrogen bonded Assemblies Based on Achiral Pyrazine-Bridged Zinc(II) Coordination Polymers with Pyrazinedione Derivatives” (Manuscript ID: chemistry-2154806).  Those comments are all valuable and very helpful for improving our paper.  We have digested these comments and made efforts to revise our manuscript following the comments.  The revised details are highlighted in red in the revised manuscript.  The reviewer’s comments are responded to one by one as follows.

Reviewer 1:

  • The authors do not provide methods of synthesis and purification of target CPs either in the article or in the Supporting Information. It is therefore not possible to assess reproducibility of the results.

Response: We are very sorry for the mistake.  We have included “Experimental Section” in the revised Supporting Information.

  • Page 4, line 160: Captions to Figure 2 and Figure 3 overlap.

Response: We are sorry for the mistake.  We have corrected them.

  • Page 6, lines 221–223: the caption to Figure 4 is located both below and above the figure.

Response: We are sorry for the mistake.  We have corrected it.

Reviewer 2 Report

This paper describes the structures of 2 similar metal complexes, one containing Zn in acentric spacegroup C2 and the other containing Co in centric spacegroup C2/m. The two structures have  similar cell dimensions and structures with differences only related to the increased symmetry in the Co complex. This unusual feature is of considerable interest.

The refinement of the two structures is mainly satisfactory. However there are no details of absorption corrections which are surely necessary. A brief experimental section should be included in the paper with details of the refinement, particularly including how the H atom positions were included and refined. It should be noted that the O-H distances are fixed via very strong constraints. Some of the H atom thermal parameters are unreasonably high and  the hydrogen atoms bonded to oxygen atoms should be given thermal parameters of 1.2 times those of the oxygen atoms to which they are bonded. Details of all hydrogen bonds should be included in the cif files. The paper should include the CCDC numbers of the structures and the appropriate reference to the CCDC.  Flack should be spelt accurately (line62).

The description of the two crystal structures needs to improved

For structure 1

The crystallographic symmetry of the positions of the two Zn atoms should b clearly stated. To say (as in line 100) that the metal atom lies in a crystallographic special position is incomplete without saying what the symmetry is.  A similar omission is to be found on line 120 for Zn(2)

Line 69. These atoms C1-C4 need to be identified else the sentence is not informative. These C atoms are not even identified in Figure S1. They should be.

Line 86 The Zn…Zn distance is surely determined accurately to 3 figures of decimals and has a standard deviation. A similar truncation to 10A in line 131 is equally unsatisfactory. In all cases the two Zn atoms need to identified as Zn1 or Zn2 together the symmetry elements.

Line 90  the distance 3.6 needs 2 more digits and a standard deviation

It is clumsy to use labels such as Zn–Npyz and Zn–OCN2pyzdione. The donor atoms are already identified as N5, N6 and O1, O2 in the text and in the tables and these numbers should be used throughout the description. Line101  The Zn atoms should be called Zn1 and Zn2 throughout the text

Line 104  The oxygen atoms involved e.g. O3, O4, O5 should be included here in the text.

In Table S2 N6 should have a symmetry element superscript equivalent to that in Figure 1. And why  use 1 and 2 superscripts in this table and I and ii superscripts in Figure 1. Only unique dimensions should be included in the table i.e. 4 bonds and 6 angles for Zn1 and 3 bonds and 3 angles for Zn2. Why is there a footnote to subscript 3? Figure S2 also uses different superscripts from Figure 1. Figure S1 (and Figure S2) should include ellipsoids for the atoms and the caption should state that hydrogen atoms are omitted. Symmetry elements should always be given in lower case. Throughout the paper superscripts for symmetry elements should be either digits 1,2…or roman numerals I, ii…not a mixture of the two types.

In Table S4 (Hydrogen bonds), it needs to be noted that as the O-H distances are fixed, all dimensions involving the H atoms should not have standard deviations included.

In my copy Figure 2 and Figure 3 are superimposed. However I can see enough of Figure 2a to establish that it is unsatisfactory and not at all clear. A better figure could surely  be obtained. Figure 2b is OK.

Quantitative details should be given of the pi…pi stacking referred to in the text.

Structure 2

Details of the structure are similar to those in structure 1 though with the increased symmetry for the C2/m spacegroup. So the symmetry around the metal atoms should be made clear.

In the description of the structure the similarity of the structure with that of 1 is not made sufficiently clear. For example there are comments about the water molecules that do not make it clear that the positions are equivalent to those found in 1.

Table S3 contains far too many dimensions. For Co1 there are only 3 unique distances and 1 unique angle (90 and 180o should not be included.) while for Co2, only 2 distances and 3 angles. The authors seem to only half-finished (or indeed only quarter-finished) inserting the hydrogen bond dimensions in Table S4. As in structure 1, only the D…A distance should be quoted with standard deviations.

Then follows a TGA study together with PXRD on 1 which shows interesting results concerning the thermal properties. However it is not made clear why any such study was not carried out on 2. The caption for Figure S5 is inadequate. The abbreviation DDSC should be explained as it is not mentioned in the main text.

Because of the interesting structural comparison of 1 and 2, the paper is of sufficient interest to  merit publication but as stated above the work requires major revision.

Author Response

Dear Dr. Yu and reviewers,

  On behalf of my co-authors, I appreciate you and the reviewers for the positive and constructive comments and suggestions on the manuscript entitled “Noncentrosymmetric Supramolecular Hydrogen bonded Assemblies Based on Achiral Pyrazine-Bridged Zinc(II) Coordination Polymers with Pyrazinedione Derivatives” (Manuscript ID: chemistry-2154806).  Those comments are all valuable and very helpful for improving our paper.  We have digested these comments and made efforts to revise our manuscript following the comments.  The revised details are highlighted in red in the revised manuscript.  The reviewer’s comments are responded to one by one as follows.

Reviewer 2

  • A brief experimental section should be included in the paper with details of the refinement, particularly including how the H atom positions were included and refined. It should be noted that the O-H distances are fixed via very strong constraints. Some of the H atom thermal parameters are unreasonably high and  the hydrogen atoms bonded to oxygen atoms should be given thermal parameters of 1.2 times those of the oxygen atoms to which they are bonded. Details of all hydrogen bonds should be included in the cif files.

Response: We included “Experimental Section” in the revised Supporting Information, where we explained synthesis, single-crystal X-ray crystallographic details, and all physical measurements.  In addition, we send the checkcif file.

  • The paper should include the CCDC numbers of the structures and the appropriate reference to the CCDC.

Response: We are sorry for the mistake.  We have corrected them.

  • Flack should be spelt accurately (line62).

Response: Thank you for pointing it out.  “Frack” has been corrected in “Flack”.

  • The crystallographic symmetry of the positions of the two Zn atoms should b clearly stated. To say (as in line 100) that the metal atom lies in a crystallographic special position is incomplete without saying what the symmetry is.  A similar omission is to be found on line 120 for Zn(2).

Response: Thank you for pointing it out.  We have numbered the two independent zinc centers to clearly distinguish them in the revised manuscript.  

  • Line 69. These atoms C1-C4 need to be identified else the sentence is not informative. These C atoms are not even identified in Figure S1. They should be.

Response: In light of the referee’s suggestion, we included the Figure S1 in the revised Supporting Information, where we numbered all related atoms and added the symmetry code.

  • Line 86 The Zn···Zn distance is surely determined accurately to 3 figures of decimals and has a standard deviation. A similar truncation to 10A in line 131 is equally unsatisfactory.  In all cases the two Zn atoms need to identified as Zn1 or Zn2 together the symmetry elements.

Response: In light of the referee’s suggestion, we have corrected them in the revised manuscript.

  • Line 90  the distance 3.6 needs 2 more digits and a standard deviation.

Response: In light of the referee’s suggestion, we have corrected it in the revised manuscript. 

  • It is clumsy to use labels such as Zn–Npyz and Zn–OCN2pyzdione. The donor atoms are already identified as N5, N6 and O1, O2 in the text and in the tables and these numbers should be used throughout the description. Line101  The Zn atoms should be called Zn1 and Zn2 throughout the text.

Response: In light of the referee’s suggestion, we have corrected them in the revised manuscript.

  • Line 104  The oxygen atoms involved e.g. O3, O4, O5 should be included here in the text.

Response: In light of the referee’s suggestion, we have corrected it in the revised manuscript.

  • In Table S2 N6 should have a symmetry element superscript equivalent to that in Figure 1. And why use 1 and 2 superscripts in this table and I and ii superscripts in Figure 1. Only unique dimensions should be included in the table i.e. 4 bonds and 6 angles for Zn1 and 3 bonds and 3 angles for Zn2. Why is there a footnote to subscript 3? Figure S2 also uses different superscripts from Figure 1. Figure S1 (and Figure S2) should include ellipsoids for the atoms and the caption should state that hydrogen atoms are omitted. Symmetry elements should always be given in lower case. Throughout the paper superscripts for symmetry elements should be either digits 1,2…or roman numerals I, ii…not a mixture of the two types.

Response: We are sorry for the confusion.  We have modified all symmetry codes appropriately.  The symbol of symmetry codes was unified i, ii ··· superscripts.

  • In Table S4 (Hydrogen bonds), it needs to be noted that as the O-H distances are fixed, all dimensions involving the H atoms should not have standard deviations included.

Response: In light of the referee’s suggestion, we corrected the Table S4 in the revised Supporting Information.

  • In my copy Figure 2 and Figure 3 are superimposed. However I can see enough of Figure 2a to establish that it is unsatisfactory and not at all clear. A better figure could surely  be obtained. Figure 2b is OK.

Response: We are sorry for the mistake.  We corrected Figures 2 and 3 in the revised manuscript.

  • Quantitative details should be given of the pi…pi stacking referred to in the text.

Response: In light of the referee’s suggestion, we have corrected it in the revised manuscript.

  • Details of the structure are similar to those in structure 1 though with the increased symmetry for the C2/m spacegroup. So the symmetry around the metal atoms should be made clear.

Response: We add the following sentence: “While 1 has long and short Zn1–N5 and Zn1–N6 bond distances, which alternaively aligned in the 1D chain, the Co1–N5 distances are symmetry related and thus they are equivalent in 2.  Reflecting the crystallographic symmetry between 1 and 2, the discrete [Co(H2O)6]2+ unit also shows more symmetric structure with two Co2–O3 and Co2–O4 bond distances of 2.063(2) Å and 2.0915(16) Å (vide supra, There are three Zn2–O3, Zn2–O4, and Zn2–O5 bonds for 1).”

  • Table S3 contains far too many dimensions. For Co1 there are only 3 unique distances and 1 unique angle (90 and 180oshould not be included.) while for Co2, only 2 distances and 3 angles. The authors seem to only half-finished (or indeed only quarter-finished) inserting the hydrogen bond dimensions in Table S4. As in structure 1, only the D…A distance should be quoted with standard deviations.

Response: In light of the referee’s suggestion, we have corrected it in the revised manuscript.

  • Then follows a TGA study together with PXRD on 1 which shows interesting results concerning the thermal properties. However it is not made clear why any such study was not carried out on 2. The caption for Figure S5 is inadequate. The abbreviation DDSC should be explained as it is not mentioned in the main text.

Response: As the focus of our communication is on properties of the chiral CP 1, not the properties of the achiral CP 2, we elected to not delve further into this point. Studies along this line of the series of CPs with other transition metal ions are now actively pursued in our lab and they will be reported in the future.  Also, we added the abbreviation of “DDSC” in the revised manuscript and supporting information.

  • Because of the interesting structural comparison of 1 and 2, the paper is of sufficient interest to merit publication but as stated above the work requires major revision.

Response: Finally, the authors thank you very much the reviewer for valuable suggestions and comments.  They wish they could acknowledge for the improvement of our manuscript.  We believe the revised manuscript is now in better shape and content and we hope the referee will be pleased with the corrections.

Round 2

Reviewer 2 Report

This paper describes the structures of 2 similar metal complexes, one containing Zn in acentric spacegroup C2 and the other containing Co in centric spacegroup C2/m. The two structures have  similar cell dimensions and structures with differences only related to the increased symmetry in the Co complex. This unusual feature is of considerable interest.

The paper has been much improved but more needs to be done

In the added experimental section details of the absorption corrections should be included.  The most important evidence for the spacegroup in 1 is provided by the Flack parameter which should be mentioned in the text.

The authors response to my request for clarification of crystallographic special positions is unsatisfactory. What is needed is to say what the symmetry is rather than quote the symmetry element as  0, y, 1/2 (Wyckoff letter b). i.e. to say the metal is on a two-fold axis

In Table S3 the angles 88.27(6) and 89.70(9) are unnecessary as they are not independent

Details of all hydrogen bonds must be included in the cif files.

What has happened to the hydrogen bond tables S4 and S5? The authors have removed all details apart from the D…A distances so that the tables are now unacceptable. In my original review I stated that only the D…A distances should have standard deviations but not that all other distances should be removed. All D-H, H…A and D-H…A dimensions must be included without standard deviations.

In  Figure S2, the bonds between oxygen atoms need to be removed.

Author Response

Dear Dr. Yu and reviewer 2,

  On behalf of my co-authors, I appreciate you and the reviewers for the positive and constructive comments and suggestions on the manuscript entitled “Noncentrosymmetric Supramolecular Hydrogen bonded Assemblies Based on Achiral Pyrazine-Bridged Zinc(II) Coordination Polymers with Pyrazinedione Derivatives” (Manuscript ID: chemistry-2154806).  Those comments are all valuable and very helpful for improving our paper.  We have digested these comments and made efforts to revise our manuscript following the comments.  The revised details are highlighted in blue in the revised manuscript.  The reviewer’s comments are responded to one by one as follows.

To reviewer 2,

  • In the added experimental section details of the absorption corrections should be included. The most important evidence for the spacegroup in 1 is provided by the Flack parameter which should be mentioned in the text.

Response: We included “Experimental Section” in the revised Supporting Information, where we explained single-crystal X-ray crystallographic details, and data analysis, data integration, and absorption collections.

  • The authors response to my request for clarification of crystallographic special positions is unsatisfactory. What is needed is to say what the symmetry is rather than quote the symmetry element as 0, y, 1/2 (Wyckoff letter b). i.e. to say the metal is on a two-fold axis

Response: Thank you for pointing it out.  We have corrected them.

  • In Table S3 the angles 88.27(6) and 89.70(9) are unnecessary as they are not independent.

Response: Thank you for pointing it out.  We have removed those lines.

  • Details of all hydrogen bonds must be included in the cif files.

Response: In light of the referee’s suggestion, we have added details of hydrogen bonding to the cif files.  

  • What has happened to the hydrogen bond tables S4 and S5? The authors have removed all details apart from the D…A distances so that the tables are now unacceptable. In my original review I stated that only the D…A distances should have standard deviations but not that all other distances should be removed. All D-H, H…A and D-H…A dimensions must be included without standard deviations.

Response: In light of the referee’s suggestion, we included the S4 and S5 in the revised Supporting Information, where we added Hydrogen-bond distances and geometries.

  • In Figure S2, the bonds between oxygen atoms need to be removed.

Response: We are sorry for the mistake.  We have corrected it.

To Dr. Yu,

1  In the acknowledgments, I forgot to mention the financial support from the university. Please allow me to add.

2  I designated Kawata and Ishikawa as corresponding authors, but I forgot to mark Ishikawa with an asterisk. Please allow me to add.